# Predicting Thermal Conductivity of Nanoparticle-Doped Cutting Fluid Oils Using Feedforward Artificial Neural Networks (FFANN)

**DOI:** 10.3390/mi16050504

**Published:** 2025-04-26

**Authors:** Beytullah Erdoğan, Abdulsamed Güneş, İrfan Kılıç, Orhan Yaman

**Affiliations:** 1Department of Mechanical Engineering, Zonguldak Bülent Ecevit University, Zonguldak 67100, Turkey; beytullah.erdogan@beun.edu.tr; 2Department of Electric and Energy, Firat University, Elazığ 23119, Turkey; 3Department of Information Technology, Firat University, Elazığ 23119, Turkey; irfankilic@firat.edu.tr; 4Department of Forensic Engineering, Firat University, Elazığ 23119, Turkey; orhanyaman@firat.edu.tr

**Keywords:** thermal conductivity, nanoparticle, feedforward artificial neural network (FFANN), cutting fluids

## Abstract

Machining processes often face challenges such as elevated temperatures and wear, which traditional cutting fluids are insufficient to address. As a result, solutions involving nanoparticle additives are being explored to enhance cooling and lubrication performance. This study investigates the effect of thermal conductivity, an important property influenced by the densities of mono and hybrid nanofluids. To this end, various nanofluids were prepared by incorporating hexagonal boron nitride (hBN), zinc oxide (ZnO), multi-walled carbon nanotubes (MWCNTs), titanium dioxide (TiO_2_), and aluminum oxide (Al_2_O_3_) nanoparticles into sunflower oil as the base fluid. Hybrid nanofluids were created by combining two nanoparticles, including ZnO + MWCNT, hBN + MWCNT, hBN + ZnO, hBN + TiO_2_, hBN + Al_2_O_3_, and TiO_2_ + Al_2_O_3_. A dataset consisting of 180 data points was generated by measuring the thermal conductivity and density of the prepared nanofluids at various temperatures (30–70 °C) in a laboratory setting. Conducting thermal conductivity measurements across different temperature ranges presents significant challenges, requiring considerable time and resources, and often resulting in high costs and potential inaccuracies. To address these issues, a feedforward artificial neural network (FFANN) method was proposed to predict thermal conductivity. Our multilayer FFANN model takes as input the temperature of the experimental environment where the measurement is made, the measured thermal conductivity of the relevant nanoparticle, and the relative density of the nanoparticle. The FFANN model predicts the thermal conductivity value linearly as output. The model demonstrated high predictive accuracy, with a reliability of R = 0.99628 and a coefficient of determination (R^2^) of 0.9999. The average mean absolute error (MAE) for all hybrid nanofluids was 0.001, and the mean squared error (MSE) was 1.76 × 10^−6^. The proposed FFANN model provides a State-of-the-Art approach for predicting thermal conductivity, offering valuable insights into selecting optimal hybrid nanofluids based on thermal conductivity values and nanoparticle density.

## 1. Introduction

During machining operations such as turning, milling, drilling, and grinding, a portion of the cutting energy is converted into heat, leading to a significant rise in temperature within the machined area. Additionally, friction at the interfaces between the tool and workpiece generates even higher temperatures in the cutting zone. This thermal buildup accelerates the wear of cutting tools, resulting in increased operational costs and a deterioration in the surface quality of the machined material. Furthermore, the elevated temperatures necessitate greater power consumption from the machine, thereby reducing overall energy efficiency. Traditionally, lubricants and coolants have been employed to mitigate these challenges. However, to address the adverse effects of elevated temperature and pressure caused by advanced machining processes, innovative solutions are required. These solutions should enhance production quality by improving the thermal conductivity of cutting fluids, achieved through the incorporation of nanoparticle additives into the base fluid, typically cutting oils, by leveraging the potential of nanotechnology.

### 1.1. Related Works

Researchers and practitioners in the machining sector have increasingly focused on enhancing production quality by improving both the cooling and lubrication effects through the addition of nanoparticles to cutting oils. Surface quality is a critical parameter that directly influences precision, which, in turn, impacts the overall cost of the product. In the study by Öndin et al., multi-walled carbon nanotubes (MWCNTs) were added to cutting fluids during stainless-steel machining. Compared to conventional MQL, the nanoparticle-enhanced fluid reduced surface roughness by 12%, cutting temperature by 38%, and tool wear by 69%, demonstrating significantly improved cooling and lubrication performance [1].

In the study conducted using CO_2_ cryogenic cooling technique in metal processing, it was stated that this method is not only an environmentally friendly application but also increases the drilling quality, tool life, and process stability by increasing the technical processing capability [2]. In addition, better surface quality and high material removal rate were obtained in machining by using nano graphene-added dielectric fluid in the processing of Ni55.8Ti shape memory alloy. It was also stated that the HTS algorithm is an effective method in optimization processes [3]. Zhu et al. used various AI models to predict the thermal conductivity of EG-Al_2_O_3_ nanofluids, with GBDT showing the best performance. SHAP analysis identified temperature as a key factor. The AI-based approach proved more accurate than traditional methods [4].

Çolak and Bayrak investigated the thermal conductivity of water-based Al_2_O_3_-Cu hybrid nanofluids using ANN and theoretical models. The Maxwell model achieved the lowest error (0.08%), while the ANN model showed high accuracy (0.4%). Results confirmed that conductivity increases with both temperature and concentration, and AI-based models offer reliable predictions [5].

ANN, LSSVM, SOM, and LM-BP algorithms were used to predict the thermal conductivity of alumina/water nanofluids based on particle size, temperature, and concentration. All models showed good predictive performance, with correlation coefficients of 0.88125 (SOM), 0.87575 (LM-BP), and 0.89999 (LSSVM), confirming their effectiveness [6]. In a related study, a genetic algorithm was employed for parameter selection and optimization within the LSSVM model, incorporating particle size in addition to nanoparticle temperature and concentration, which are commonly considered in thermal conductivity modeling. For this model, which predicted thermal conductivity by using nanoparticle size as an input variable, the coefficient of determination (R^2^) was 0.9902, and the mean squared error for the thermal conductivity ratio of Al_2_O_3_/EG was found to be 8.64 × 10^−4^ [7]. Another study involving Al_2_O_3_/EG nanofluids reported that the ANN model accurately predicted experimental thermal conductivity data with a high regression coefficient (R = 0.9993). In contrast, the Hamilton–Crosser and Lu–Lin models failed to predict thermal conductivity values across varying temperatures and concentrations. A new correlation equation was developed using experimental data, and this equation, along with the ANN model, successfully predicted the thermal conductivity [8].

An artificial neural network (ANN) was used to predict the thermal conductivity rate in experiments using SWCNT-CuO/water nanofluid in the range of 28–50 °C and at volumetric concentrations of 0.03–1.15%. MLP network with Lundberg–Marquardt algorithm (LMA) was used by ANN to predict the data. The R^2^ and MSE of the optimum choices were 0.9999029 and 6.33377 × 10^−6^, respectively. Experiments compared the correlation and ANN efficiencies, showing that the ANN method gives more accurate predictions [9].

The study investigated the viscosity and thermal conductivity properties of water-based nanofluids using nanoparticle-doped cutting fluid, using Al_2_O_3_ nanoparticles. In this study, it was observed that the thermal conductivity of Al_2_O_3_–water nanofluid increases in direct proportion to the concentration. The study showed that low concentrations of nanoparticles provide a positive effect, especially on thermal conductivity [10].

In the study on the machining of AISI 4340 hardened steel, Bag et al. investigated the potential of nanofluids to enhance the thermal conductivity of cutting fluids. The authors highlight that nanoparticles such as Al_2_O_3_, with a high heat dissipation capacity, can significantly improve the cooling and lubrication properties, as well as the overall machining performance. The study underscores the importance of nanofluid stability in optimizing the thermal conductivity of cutting fluids and suggests that further experimental investigations at varying concentration ratios are necessary to fully understand the effects [11].

Ravi et al.’s experimental study demonstrates the benefits of incorporating nanofluids into metalworking processes. The study reveals that the use of nanofluids as lubricants positively influences key machining parameters, particularly by enhancing viscosity and thermal conductivity—factors that play a crucial role in improving machining efficiency [12].

Gajrani et al. developed a hybrid nano-green cutting fluid (HN-GCF) using CaF_2_ and MoS_2_ nanoparticles in mineral oil. Applied with minimum quantity cutting fluid (MQCF) techniques, HN-GCF improved thermal conductivity and enhanced workpiece surface quality by 37%, outperforming conventional lubricants [13].

Selvarajoo et al. investigated the thermal conductivity of nanofluid combinations containing Al_2_O_3_ and graphene oxide (GO) in the temperature range of 30–50 °C. The results showed a 4.30% increase in thermal conductivity for Al_2_O_3_ and 4.34% for GO at 1% volumetric concentration of the hybrid nanofluid. The study achieved a margin of error of 4–6% in its estimates [14].

It has been stated that the use of cryogenic LCO_2_ in the machining of Ti-6Al-4V alloy can be used as the most suitable cutting fluid for nanoparticles in terms of both machining performance and environmentally friendly sustainability [15].

Sharma et al. studied the effects of molybdenum disulfide (MoS_2_) and alumina nanoparticles on the thermal conductivity and viscosity of cutting fluid in turning operations on AISI 304 steel sheets. Regression models were used to estimate the unmeasured parameters. An increase in thermal conductivity from about 71% to 152% was observed [16].

In their studies on adding nanoparticles to vegetable oils for use in processing applications, Yadav et al. used a 60–40% Al_2_O_3_–graphene hybrid nanofluid in sunflower oil as the base fluid, especially during turning operations. They found that the hybrid nanofluid reduced the cutting force by 7.6% compared to the single nanofluid, which indirectly provided an increase in viscosity, which improved the thermal conductivity [17].

Hirudayanathan et al. investigated the effect of particle size, shape, and material type on the thermal conductivity and viscosity of nanofluids. While ceramic nanoparticles provide high stability, metallic and carbon-based nanoparticles are more likely to deposit and reduce machining performance. In addition, the high interaction of nanoparticles with the base fluid in metal cutting fluids can cause surface roughness problems due to agglomeration [18].

Manikanta et al. found that antifriction, thermal conductivity, and the cooling rate increased with increasing volumetric concentration ratios of nanoparticles added to base fluids in cutting fluids. In addition, it was reported that surface roughness and cutting forces decreased significantly with increasing tribological properties [19].

Duc et al. investigated the use of Al_2_O_3_ nanofluid in cutting fluids for drilling operations on Hardox 500 steels. Their findings revealed that the use of this nanofluid resulted in higher viscosity, improved heat transfer, enhanced surface finish, and reduced cutting force compared to pure water [20].

Arifuddin et al. employed a hybrid nanofluid consisting of Al_2_O_3_ and TiO_2_ nanoparticles in high-speed machining fluids. Minimum quantity lubrication (MQL) was utilized to assess surface roughness, cutting temperature, and tool heating. The Al_2_O_3_-TiO_2_ nanofluid, used at a 4% volumetric concentration, exhibited the highest thermal conductivity and the lowest surface roughness ratio [21].

In a study by Singh et al., the positive effects of adding nanofluids to cutting fluids were observed across various metalworking processes, including drilling, grinding, milling, and turning. The investigation determined that the increase in thermal conductivity of cutting fluids enhanced by nanoparticles was a crucial factor in improving machining performance. While the impact varied across different processes, the use of nanofluids significantly improved dimensional accuracy and reduced cutting temperature, cutting forces, tool wear, and the friction coefficient. The study reported positive results from the use of nanofluids containing nanoparticles such as MoS_2_, TiO_2_, Al_2_O_3_, and SiO_2_, at concentrations ranging from 0.2% to 2%, in turning, milling, drilling, and grinding operations [22]. However, the study also highlighted the absence of scientific methodologies to identify the most suitable nanofluid for use as a cutting fluid in specific metals and alloys with optimal performance.

Prabhu et al. developed a nanofluid by incorporating multi-walled carbon nanotube (MWCNT) nanoparticles, with sizes ranging from 10 to 20 nm, at a 0.02% volumetric concentration into SAE20W40 cutting fluid. The surface roughness values resulting from the application of this cutting fluid were compared using an artificial neural network (ANN) methodology, which applied the comparison across various variables. The fuzzy logic model developed in this study yielded low prediction errors, ranging from 0.88% to 9.23% [23].

Vignesh et al. prepared a hybrid nanofluid by adding TiO_2_, ZnO, and Fe_2_O_3_ nanoparticles to coconut oil as the cutting fluid. The use of this nanofluid resulted in improved performance in cutting forces, tool wear, and surface roughness, attributed to enhanced lubrication and cooling properties and increased heat transfer [24].

In the machining of EN-31 steel, hexagonal boron nitride (hBN) nanoparticles were incorporated into the cutting fluid to enhance the lubrication and cooling effects within the minimum quantity lubrication (MQL) system. A study utilizing Gray Relational Analysis (GRA) identified that the lubricant flow rate parameter had the highest significance in the multiple response factor, contributing a value of 558.1 [25].

In a study where high-speed milling was performed using TiAlN-coated carbide end mills, dimensional errors and surface roughness were investigated. It was stated that tool deflection was more pronounced on low inclined surfaces and small-diameter tools [26]. In addition, dimensional errors in ball nose-milling operations in a similar machining operation were estimated using ANN. It was stated that RBF models had better prediction success (RMSE, 1.83 µm; correlation coefficient, 0.897) in all scenarios [27]. In another study, the thermal properties of TiO_2_ nanofluid were analyzed by employing artificial neural networks (ANNs) to predict the thermal conductivity and viscosity characteristics of the nanofluid. The modeling results indicated that the ANN model accurately predicted thermal conductivity, with the best model yielding a mean squared error (MSE) of 4.2484 × 10^−6^ and an R^2^ value of 0.99982. The developed correlations demonstrated a high level of agreement with the experimental data, with a deviation of ±3.5% for thermal conductivity [28].

In the study where four different vegetable oils (high-oleic sunflower oil, regular sunflower oil, castor oil, and recycled ECO-350 oil) were used instead of canola oil for machining Inconel 718 alloy using minimum quantity lubrication technology (MQL), it was found that high-oleic sunflower oil provided 15% longer tool life [29].

Metals exhibit higher thermal conductivity compared to organic materials and liquids. Consequently, incorporating metal or metal oxide nanoparticles into a base fluid enhances the heat-carrying capacity of the resulting mixture. Previous studies have focused on nanoparticles of millimeter or micrometer size, which exhibit low stability within the liquid medium. However, suspending solid particles smaller than 100 nm in the base fluid offers superior heat transfer properties owing to their enhanced thermal conductivity and improved stability relative to the base liquid [30]. The effect of the use of nanofluids on heat transfer was investigated in the study of B. Erdoğan et al. [31].

TiO_2_-Anatase, TiO_2_-Rutile, SnO_2_, Co_3_O_4_, CuO, ZnO, and Al_2_O_3_ nanoparticles were used in base liquids as ethylene glycol, water, polyethylene glycol, ethylene glycol + water, and polyethylene glycol + water, and the density and isothermal compressibility properties were estimated using the hard-chain equation of state (EoS). In the study, the trained neural network obtained high-accuracy results in estimating the densities of nanofluids with three parameters, such as temperature, pressure, and nanoparticle mole fraction [32].

### 1.2. Motivation and Contributions

Despite the known positive impact of nanoparticle usage in the machining sector, it has been observed that there is currently no systematic process to provide a balanced cost–performance relationship by determining the optimum nanofluid type, concentration, and thermal conductivity values in various metalworking processes. This study aims to increase efficiency, reduce costs, and save time in metalworking operations by determining the optimum combinations that balance maximum cooling performance with minimum cost or both. One of the key factors in selecting the most effective coolant in the use of single, hybrid, and ternary nanoparticles in cooling oils is to give priority to nanofluids with higher thermal conductivity values.

### 1.3. Organization of the Study

In the Introduction section of the study, recent research on thermal conductivity is reviewed, outlining the rationale for the study and its contributions. The Materials and Methods section details the preparation of the nanofluids and the measurement of their thermal conductivity, followed by a presentation of the dataset generated from these measurements. The proposed methodology, which outlines how the dataset is trained using the artificial neural network, is described in detail. In the Experimental Results section, the outcomes for all hybrid nanofluids, as obtained through the proposed method, are presented with comprehensive graphics and tables. The Discussion section compares the obtained results with those of similar studies conducted in recent years, highlighting the advantages and limitations of the proposed method. Finally, in the Conclusion section, the experimental findings and broader results, as discussed, are summarized.

## 2. Nanofluid Material Preparation, Thermal Conductivity, and Dataset

### 2.1. Nanofluid Preparation

In the experiments, sunflower oil, a sustainable and widely adopted base fluid in industrial applications, was chosen for its ability to maintain stability at the tool–chip interface during machining, especially under challenging conditions. Additionally, sunflower oils enhanced with specific additives for machining processes have demonstrated significant improvements in machining performance [33]. In this study, the specially formulated sunflower oils tailored for machining were utilized, and their detailed technical specifications are provided in Table 1.

In the preparation of the nanofluids, five distinct nanoparticles were incorporated into the base fluid, both in mono and hybrid forms. The nanoparticles selected for this study—hexagonal boron nitride (hBN), zinc oxide (ZnO), multi-walled carbon nanotubes (MWCNTs), titanium dioxide (TiO_2_), and aluminum oxide (Al_2_O_3_)—were chosen due to their high viscosity and thermal conductivity characteristics. Nanoparticles are much more reactive and more accessible to exposure than their macroscopic forms, so they should be used with caution for the health of users and workers. Long-term respiratory exposure can have dangerous consequences. TiO_2_ and MWCNT nanoparticles in particular are among the particles that have carcinogenic effects. The properties of these nanoparticles are presented in Table 2.

A total of 11 distinct nanofluids, consisting of 5 mono and 6 hybrid formulations, were prepared using a two-stage method. In the first stage of this method, nanoparticles are produced as dry powders through physical or chemical processes. The second stage involves dispersing the produced nanoparticles into the base fluid. In the mono nanofluids, each nanoparticle listed in Table 2 was used individually at a volumetric concentration of 0.5%. For the hybrid nanofluids, which consist of two types of nanoparticles, each type was added in equal proportions (0.25%) to achieve a total volumetric concentration of 0.5%. The hybrid nanofluids prepared included ZnO + MWCNT, hBN + MWCNT, hBN + ZnO, hBN + TiO_2_, hBN + Al_2_O_3_, and TiO_2_+. In the case of ternary hybrid nanofluids, where three types of nanoparticles were utilized, the volumetric concentration was again 0.5%, with each nanoparticle type contributing 0.17%. Al_2_O_3_. The mass of each nanoparticle was calculated based on the volumetric concentrations (0.5% for mono nanofluids and 0.25% for hybrid formulations), using the conversion formulas outlined in Table 3 [33]. The calculated mass of each nanoparticle was then measured on a precision balance and transferred into separate containers for further processing (Figure 1).

During the preparation of nanofluids, nanoparticles were accurately weighed using a precision scale and then combined with vegetable oil, which served as the base fluid. The initial mixing was performed using a magnetic stirrer for 15 min. Following this physical mixing, the resulting mixture was subjected to ultrasonic homogenization (brand/model: Optical Ivy System CY—500; power = 500 W; frequency = 20 kHz, and probe diameter/length = Ø5, 6/60 mm) for a minimum of 30 min to achieve a homogeneous dispersion. The procedural steps involved in the preparation of the cutting fluids are illustrated both experimentally and schematically in Figure 2.

### 2.2. Thermal Conductivity and Its Measurements

Thermal conductivity is a property that defines the ability of materials to conduct heat and is usually indicated by the letter “k”. Thermal conductivity is related to the amount of heat (Q), the thickness of the material (L), unit time (t), the surface area where heat transfer occurs (A), and the temperature difference that causes heat transfer (ΔT). Under steady-state conditions and when heat transfer depends only on the temperature gradient, this property can be calculated with Equation (1) [34,35,36].(1)$$k=\frac{Q}{t}\times \frac{L}{A\times ∆T}$$

Thermal conductivity measurements of all nanofluids (KD2 Decagon) were carried out using a thermal conductivity measuring device with a measurement range of 0.02–2.00 W/m·K.

### 2.3. Dataset

As depicted in Figure 3, a total of 30 distinct thermal conductivity values were measured for each fluid across a temperature range of 30–70 °C for both single and hybrid nanofluid mixtures. The thermal conductivity values obtained at different temperatures for the base fluid oil, five mono nanofluids, and six hybrid nanofluids are presented in Table 4 and Table 5. Upon examining the values in Table 4 and Table 5, it is evident that the temperature values are not arranged in a sequential order (e.g., 30.00, 30.10, 30.20, 30.30, etc.). This irregularity arises from the challenges associated with measuring all values within the specified range in a systematic manner, considering both time constraints and cost implications. Therefore, it is more efficient to estimate the thermal conductivity values within a defined range using a high-accuracy method based on the available measurements. This approach ensures a more consistent analysis by increasing the number of measurements with enhanced precision.

## 3. Methodology and Proposed Method

The method involves estimating the thermal conductivities of nanofluids measured at various temperatures under laboratory conditions by recalculating these values based on the densities of the fluids. The detailed methodology employed is illustrated in Figure 4. A step-by-step implementation of this methodology is presented in Algorithm 1. By utilizing the thermal conductivity (TC_1_) and thermal conductivity (TC_2_) values of Nanofluid 1, calculated for each measured temperature, the data were trained in a feedforward artificial neural network, as shown in Figure 5, with an 80% training and 20% testing split, incorporating a total of three inputs (T, TC_1_, and TC_2_). The hybrid thermal conductivity of the values measured in the laboratory is determined by calculating the weights (w_1_ and w_2_), as outlined in Equations (2) and (3), based on the concentrations (ρ) of the nanofluids provided in Table 2. The weighted average hybrid thermal conductivity (hTC) is then computed as specified in Equation (4). The dataset is divided into 80% training and 20% test data. Training data were randomly selected from the data pool to obtain consistent results. Only 80% of the data selected for training were randomly selected from 180 data. The remaining 20% of the data were used for testing and were definitely not used for training the model. Validation data were not used because the number of data was very small. In order to verify the accuracy of the training data selection, our model was also trained using k-fold cross-validation. Due to the limited measured data due to the difficulties in the experimental environment, the k value was given as 5, and the model was trained in 20% pieces (each piece is 80% training and 20% test). In the study conducted by Claudio A. Faúndez et al., the authors examined how to eliminate misleadings in the estimation of fluid properties using artificial neural Network and how to conduct an accurate and consistent analysis [37].
**Algorithm 1.** Proposed method’s algorithm for thermal conductivity estimation (for each hybrid nanofluid)**Input:** *T*, *TC*_1_, *TC*_2_, *ρ*, and hybrid1_datas**Output:** *pTC*, *MAE*, *MSE*, *MAPE*, *R*^2^**1**size ← Load(hybrid1_datas)**2**i ← 0**3****if** i < 30 **then****3.1**   wi1=ρ1ρ1+ρ2, *w_i_*_2_=ρ2ρ1+ρ2, *hTC_i_* = wi1∗TCi1+wi2∗TCi2
**3.2**   inputDatas(***T_i_, TC_i_*_1_*, TC_i_*_2_**)**3.3**   outputDatas(***hTC_i_***)**3.4**   i = i + 1, **go to step 3****4**trainRatio = 0.8 trainIndex = randperm(size, round(trainRatio * size))XTrain = inputDatas(trainIndex, :)’, YTrain = outputDatas(trainIndex, :)’testIndex = setdiff(1:size, trainIndex)XTest = inputDatas(testIndex, :)’, YTest = outputDatas(testIndex, :)’net = feedforwardnet([10 10])net.trainFcn = ‘trainbr’net.layers{1}.transferFcn = ‘tansig’net.layers{2}.transferFcn = ‘tansig’net.layers{3}.transferFcn = ‘purelin’net.trainParam.epochs = 1000net.trainParam.goal = 10^−1000^**5****net** = train(net, **XTrain**, **YTrain**)**6*****pTC*** = net(XTest)**7***MAE* = mean(abs(*pTC* − YTest))*MSE* = mean((*pTC* − YTest).^2);*MAPE* = mean(abs((*pTC* − YTest)/YTest)) * 100*R*^2^ = 1 − mean(((*pTC*-YTest).^2)/(YTest.^2))**8****return***pTC*, *MAE*, *MSE*, *MAPE*, *R*^2^


(2)
$$w_{1}=\frac{\rho_{1}}{\rho_{1}+\rho_{2}}$$



(3)
$$w_{2}=\frac{\rho_{2}}{\rho_{1}+\rho_{2}}$$



(4)
$$hTC=w_{1}*{TC}_{1}+w_{2}*{TC}_{2}$$


By changing the parameters of our artificial neural network model, a model that produces 3 inputs, 2 × 10 neurons, 1 × 1-cell hidden layer, and 1 output feedforward linear output was decided (Figure 5). The reason why our proposed artificial neural network model was selected for different scenarios is covered in the Discussion section based on the results obtained according to different scenarios in Table 10. When the results in Table 10 are evaluated, it is seen that the results of the model we proposed in two-hidden-layer models are better or the same in terms of all metrics. Only for the test data, the MAE value was 0.00048 for the [20 20] layer, while it was 0.0005 for the [10 10] layer (which we proposed). Among the three-hidden-layer models, our model gives better results for all metrics except the [5 5 5]-layer model MAE value (0.00090) for the training data. It was also seen that the result was best with Bayesian optimization for regularization and optimization (Table 6).

Bayesian regularization was applied to the feedforward neural network for the regularization process, as depicted in Figure 5. The hyperbolic tangent (tanh) activation function was employed for both the first and second hidden layers, while the linear transfer function (purelin) was used for the final hidden layer. The performance of the model was evaluated by calculating various metrics, including mean absolute error (MAE), mean squared error (MSE), mean absolute percentage error (MAPE), correlation coefficient (R), and coefficient of determination (R^2^), based on the thermal conductivity (pTC) values predicted by the model and the concentration-dependent average hybrid conductivity (hTC) values. These evaluation metrics are defined in Equations (5)–(9), respectively.(5)$$MAE=\frac{1}{n}\sum_{i=1}^{n}\left|{pTC}_{i}-{hTC}_{i}\right|$$(6)$$MSE=\frac{1}{n}\sum_{i=1}^{n}{\left({pTC}_{i}-{hTC}_{i}\right)}^{2}$$(7)$$MAPE=\frac{1}{n}\sum_{i=1}^{n}\left|\frac{{pTC}_{i}-{hTC}_{i}}{{hTC}_{i}}\right|\times 100$$(8)$$R=\frac{\sum_{i=1}^{n}{pTC}_{i}\times {hTC}_{i}-\sum_{i=1}^{n}{pTC}_{i}\times \sum_{i=1}^{n}{hTC}_{i}}{\sqrt{\sum_{i=1}^{n}{{hTC}_{i}}^{2}-{\left(\sum_{i=1}^{n}{hTC}_{i}\right)}^{2}}\sqrt{\sum_{i=1}^{n}{{pTC}_{i}}^{2}-{\left(\sum_{i=1}^{n}{pTC}_{i}\right)}^{2}}}$$(9)$$R^{2}=1-\frac{\sum_{i=1}^{n}{\left({pTC}_{i}-{hTC}_{i}\right)}^{2}}{\sum_{1=1}^{n}{\left({pTC}_{i}\right)}^{2}}$$

When Table 6 is examined, it is seen that very fast results are obtained for 1000 epochs (10 s).

Table 6 shows some important parameter values during the training process while training the FeedForwardNet network. These parameters are defined as follows:

**Gradient:** It expresses the magnitude of the gradient (derivative) value calculated over the weights and biases. It is taken into account when determining how much the Bayesian optimization algorithm should change the weights.

**Mu:** It is a damping parameter used in the Bayesian algorithm. If Mu is large, the algorithm works like gradient descent. If Mu is small, the algorithm works similarly to the Gauss–Newton approach. If the error decreases as the training progresses, Mu decreases; if the error increases, Mu increases.

**Effective # Parameters:** It is the number of effective parameters. It expresses the number of parameters that actually contribute to the learning of the network. It is usually important when regularization is used.

**Sum Squared Parameters:** It is the sum of the squares of the weight and bias values of the network. This value is usually used to monitor the size of the weights.

## 4. Experimental Results

The software implementation of the proposed methodology was carried out using the MATLAB 2023A IDE on a computer equipped with 16 GB of memory and an Intel i7 9th Generation processor. Figure 6 illustrates the comparison between the thermal conductivity values predicted by the proposed feedforward deep learning model and the experimentally measured values. The results demonstrate that the model’s predictions are in close agreement with the measured data. The overall performance of the model is summarized in Table 7. As can be seen in Table 7 and Figure 7, the performance results of our model for the training data are better than the performance results for the test data, as expected. 

Table 7 also provides training and test results for five parts (20%), using k = 5 for k-fold cross-validation. When the results are examined, it is seen that the results are very close or slightly lower than the general results. This confirms that the results obtained from the data selected for training (80%) are consistent and reliable.

As seen in Figure 7, the smallest mean squared error (MSE) was obtained in the 880th epoch (889th epoch is written in the graph since it started from the zeroth epoch). The MSE values measured for train and test are very close to each other. This shows that our model is consistent.

The lowest gradient, Mu, gamk, ssX, and valfail change graphs obtained at the end of model training are given in Figure 8. As seen in the graphs, our model achieved the best result in epoch 880.

Figure 9 shows the correlation coefficients (R) graphs for training, test, and all data (training and test) as a result of training the model. The correlation coefficients were obtained as R = 0.99599 for training data, R = 0.99783 for test data, and R = 0.99628 for all data. The fact that the correlation coefficients are very close to 1 (100%) proves the reliability of the predictions made by our model. Residual plots for training and testing are given in Figure 10. When the residual plots are examined, it is seen that the narrow range for training and testing is [−2 × 10^−3^,+2 × 10^−3^]. The wide range for training is [−6 × 10^−3^,+4 × 10^−3^], and for testing, it is [−5 × 10^−3^,+4 × 10^−3^]. It is seen that the residual distribution is more balanced when the thermal conductivity is 0.17 and above for training and testing data.

Figure 11 presents the comparison between the measured and model-predicted thermal conductivity values for the ZnO + MWCNT, hBN + MWCNT, hBN + ZnO, hBN + TiO_2_, hBN + Al_2_O_3_, and TiO_2_ + Al_2_O_3_ hybrid nanofluids. The graphs reveal a strong correlation between the measured and predicted values, with particularly close overlap observed in the hBN + ZnO, hBN + TiO_2_, and TiO_2_ + Al_2_O_3_ hybrid mixtures. These results demonstrate the accuracy of the model’s predictions. Table 8 displays the MAE, MSE, and MAPE values obtained after model training, while Table 9 provides the statistical error rates. Upon reviewing Table 9, it is evident that the hBN + ZnO and hBN + TiO_2_ hybrid mixtures exhibit the lowest error, particularly in terms of average error. In the plots in Figure 11 and Figure 12, the X-axis represents all measured values (30 data) of each hybrid nanofluid given as samples. These samples include all of the training and test data. The statistical error values given in Figure 13 and Figure 14 and Table 9 were calculated according to Equation (10).(10)$$Error\;rate=\frac{1}{n}\sum_{i=1}^{n}{|TCPred}_{i}-{TCmeasured}_{i}|*100$$

Residual plots of each hybrid nanofluid mixture are given in Figure 12. When the six residual plots are examined, it is seen that the residual values move away from the average value at high conductivity. The residual value ranges are (−5 × 10^−3^; −5 × 10^−2^), (−3 × 10^−2^; 1 × 10^−2^), (−1 × 10^−2^; 1.5 × 10^−2^), (−5 × 10^−3^; 2.5 × 10^−2^), (−2.5 × 10^−2^; 1 × 10^−2^), and (−2.5 × 10^−2^; 1.5 × 10^−2^), respectively. It is seen that the residual distribution is more balanced for hBN + ZnO and hBN + TiO_2_ mixtures, and this situation is not affected much at high conductivity. It is seen that high conductivity affects the residual distribution negatively for hBN + Al_2_O_3_ and TiO_2_+ Al_2_O_3_ mixtures.

In addition, it is seen in Table 9 that the maximum error rate (max) and standard deviation (Std) values of the ZnO + MWCNT and hBN + MWCNT hybrid mixtures are very high. Therefore, it can be said that these mixtures cannot provide good performance in terms of thermal conductivity. Similarly, this situation is confirmed for these mixtures in the residual plots.

Figure 13 illustrates the error rate plots for the statistical error rates of the six hybrid mixtures. As shown in Table 9, the lowest error values are observed in the hBN + ZnO and hBN + TiO_2_ hybrid mixtures. Figure 14 presents the comparison between the measured (actual) thermal conductivity values and the model-predicted thermal conductivity values for the six hybrid mixtures. The actual thermal conductivity values are in the range of [0.13–0.18], while the predicted values from our model fall within the range of [0.13–0.20]. Upon examining the plots, it is evident that the hBN + ZnO, hBN + TiO_2_, and hBN + Al_2_O_3_ hybrid mixtures display a high degree of consistency, with the ZnO + MWCNT hybrid mixture showing an acceptable level of deviation.

## 5. Discussion and Comparison with Other Studies

The artificial neural network model we proposed has been shown to be the most reasonable solution in terms of performance metrics according to different scenarios (number of hidden layers and number of neural network cells) given in Table 10. As can be seen in Table 10, it is clear why our proposed model was chosen among the two-hidden-layer neural model and three-hidden-layer neural network models.

Although no study directly identical to ours exists in the literature, there are several similar investigations regarding thermal conductivity with various nano-hybrid mixtures. Table 11 presents a comparative analysis of the results obtained in our study alongside those from similar studies. It is important to note that due to the use of different materials, a direct comparison of the performance metrics may not be entirely appropriate. However, a general evaluation can be made to provide an overarching perspective.

As shown in Table 11, recent studies typically employ two or three hybrid nano-mixtures. In the study by Yunyan Shang et al., Grid Search (GS), Random Search (RS), and Bayesian optimization, in conjunction with a multilayer perceptron neural network (MLPNN), were used, with the best results obtained using the GS-MLPNN method. In Mohammadreza B. et al.’s research, the Group Method of Data Handling (GMDH) combined with the Non-Dominated Sorted Genetic Algorithm-II (NSGA-II) metaheuristic optimization method produced the most favorable results, representing the latest advancements in the field. P. K. Kanti et al. achieved the best performance using the Random Forest algorithm. Meanwhile, M. Dinesh Babu et al. investigated the thermal conductivity of Al_2_O_3_CuO/water hybrid nanofluids using an artificial neural network (ANN) optimized by the Levenberg–Marquardt method. Shekhar et al. focused on the thermal conductivity of Al_2_O_3_, CeO_2_, and CuO nanofluids, while Fevzi Şahin’s study yielded high-accuracy results with Levenberg–Marquardt optimization and a multilayer perceptron (MLP) network. Our study stands out for incorporating a larger variety of mono and hybrid nanofluids in comparison to these previous studies.

## 6. Conclusions

In conclusion, this study investigated the thermal conductivities of hybrid nanofluids based on relative concentrations for five different mono nanofluids and six distinct hybrid nanofluids (ZnO + MWCNT, hBN + MWCNT, hBN + ZnO, hBN + TiO_2_, hBN + Al_2_O_3_, and TiO_2_ + Al_2_O_3_). Using the proposed Bayesian-optimized feedforward artificial neural network (FFANN) deep learning model, the results obtained were in close alignment with State-of-the-Art findings. The overall mean squared error (MSE) was measured as 1.76 × 10^−6^, demonstrating the model’s high accuracy. Additionally, the model achieved an overall mean absolute percentage error (MAPE) of 0.1490, a noteworthy result compared to existing studies in the literature. The results were obtained within a stability range of R^2^ = 0.9999, indicating that the relative concentration approach, coupled with the artificial intelligence-based network, can accurately predict thermal conductivity in real time.

The high correlation coefficients (R > 0.99) and the narrow, balanced residual distributions demonstrate the accuracy and reliability of the model’s predictions, particularly for thermal conductivity values of 0.17 and above (in Figure 9 and Figure 10).

The close agreement between measured and predicted thermal conductivity values, especially for the hBN + ZnO and hBN + TiO_2_ hybrid nanofluids, along with low statistical error rates, confirms the high accuracy and reliability of the developed model (in Figure 11).

The comparison of actual and predicted thermal conductivity values demonstrates that the model provides highly consistent results for hBN + ZnO, hBN + TiO_2_, and hBN + Al_2_O_3_ hybrid mixtures, with the lowest error rates observed for hBN + ZnO and hBN + TiO_2_, confirming the model’s robustness and predictive accuracy (in Figure 12 and Figure 13).

Furthermore, the model successfully predicted the thermal conductivities of hybrid nanofluids (ZnO + TiO_2_, ZnO + Al_2_O_3_, MWCNT + TiO_2_, and MWCNT + Al_2_O_3_), for which no experimental data were available. The predictions for these four hybrid nanofluids were made with a high degree of reliability (R^2^ = 0.9803). This approach demonstrates the potential to make precise predictions without the need for costly and complex laboratory setups.

Future investigations may focus on the implementation of multi-objective metaheuristic approaches to further refine and optimize the accuracy and performance of the proposed model. Moreover, the design and development of a practical measurement device prototype, augmented with artificial intelligence, could provide a more efficient, cost-effective, and accessible method for real-time thermal conductivity measurements.

## Figures and Tables

**Figure 1 micromachines-16-00504-f001:**
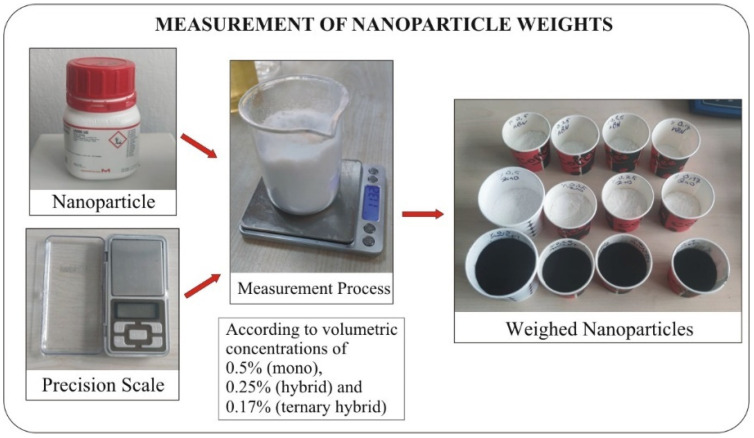
Measurement of particle weights according to masses corresponding to volumetric concentrations.

**Figure 2 micromachines-16-00504-f002:**
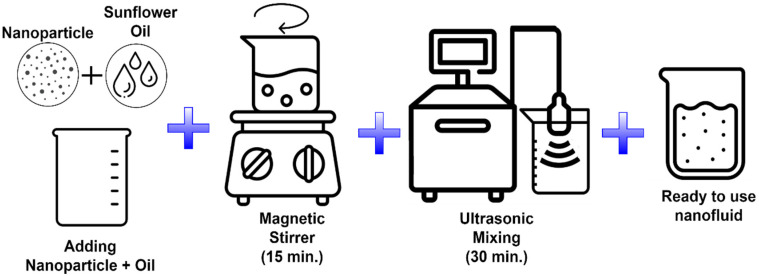
Steps applied in the preparation of nanofluids.

**Figure 3 micromachines-16-00504-f003:**
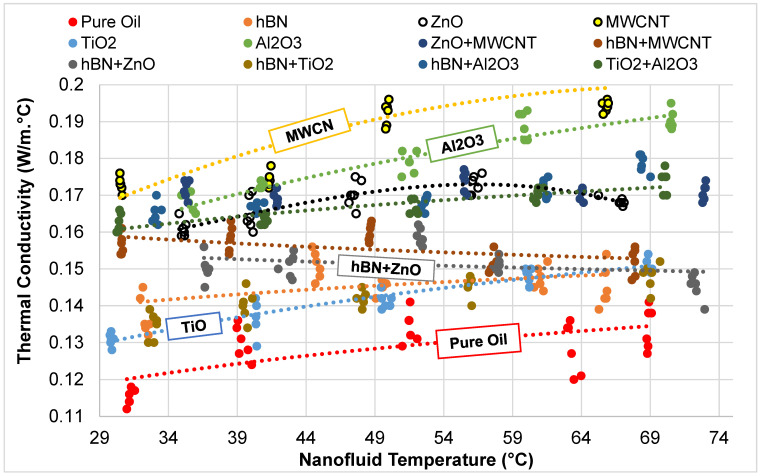
Measured thermal conductivity values of nanofluids at different temperatures.

**Figure 4 micromachines-16-00504-f004:**
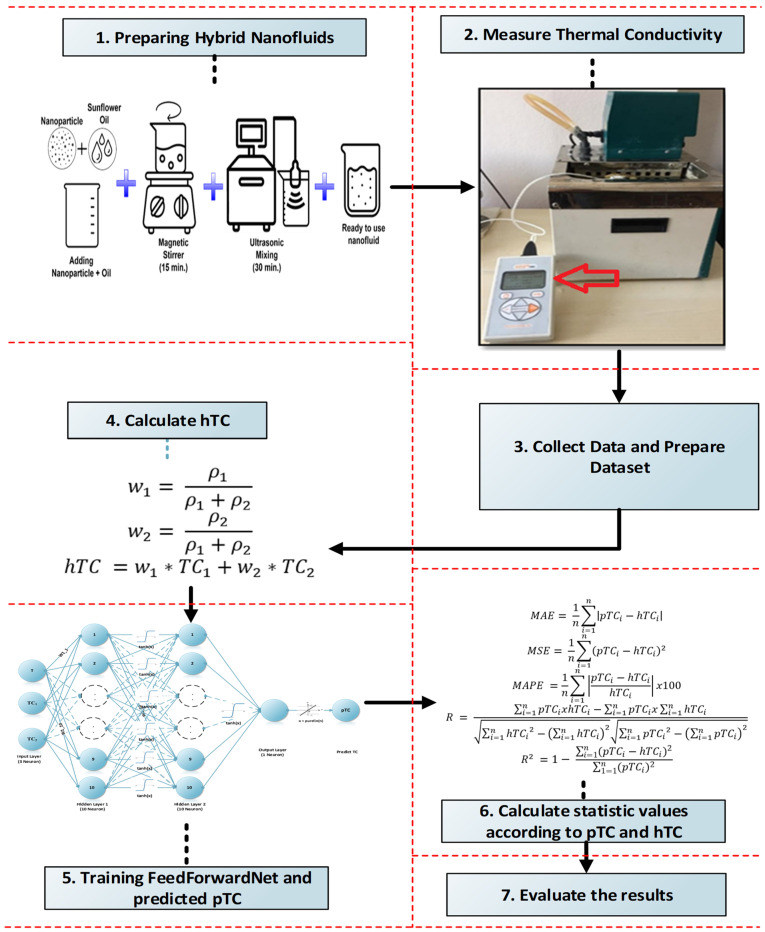
Methodology of the study.

**Figure 5 micromachines-16-00504-f005:**
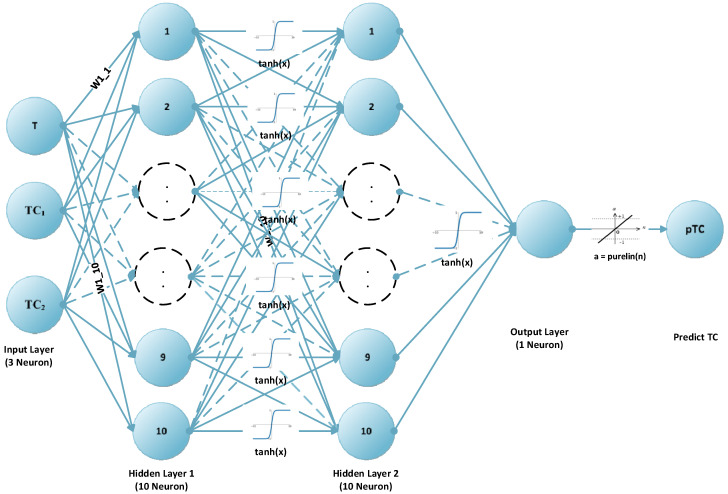
Proposed feedforward neural network model (FFANN).

**Figure 6 micromachines-16-00504-f006:**
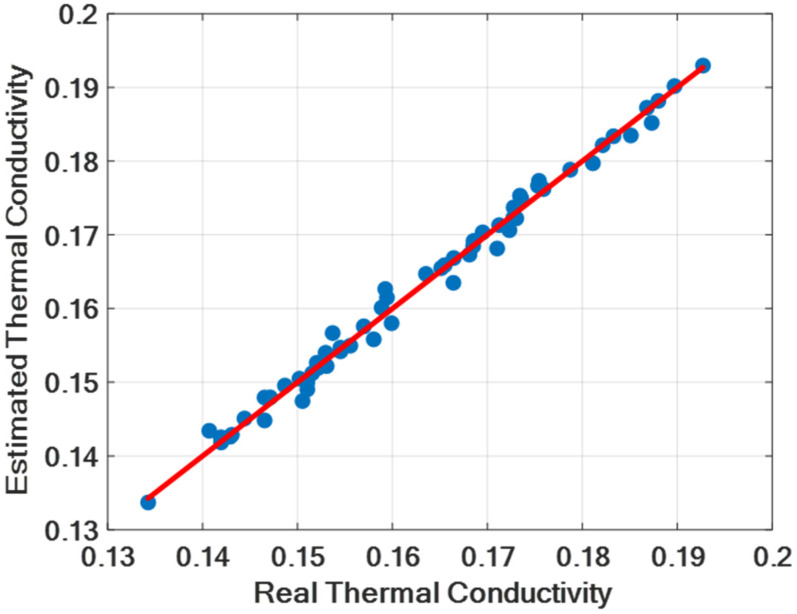
Thermal conductivity predictions of our deep learning model based on real (measured) values.

**Figure 7 micromachines-16-00504-f007:**
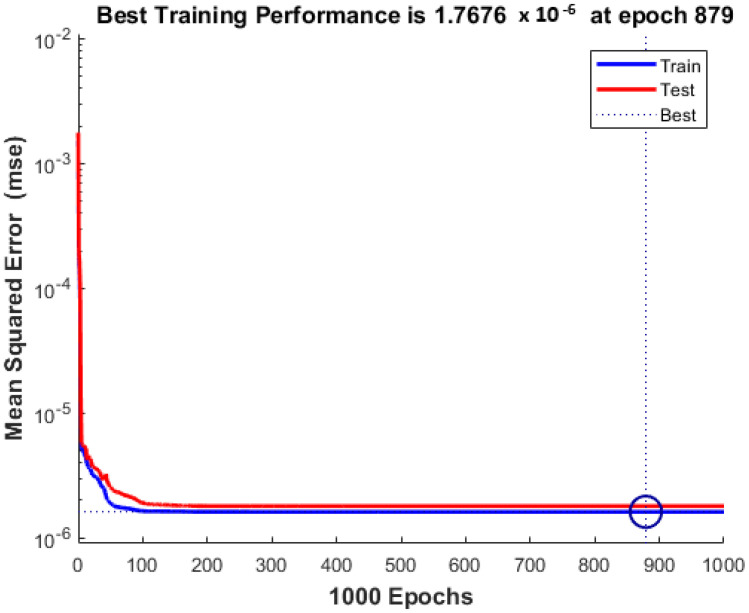
MSE plot obtained as a result of model training.

**Figure 8 micromachines-16-00504-f008:**
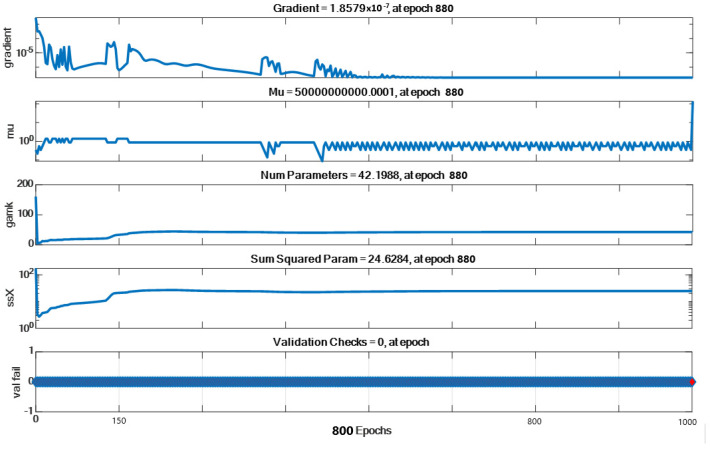
Gradient, Mu, gamk, ssX, and valfail plots at the end of our model training.

**Figure 9 micromachines-16-00504-f009:**
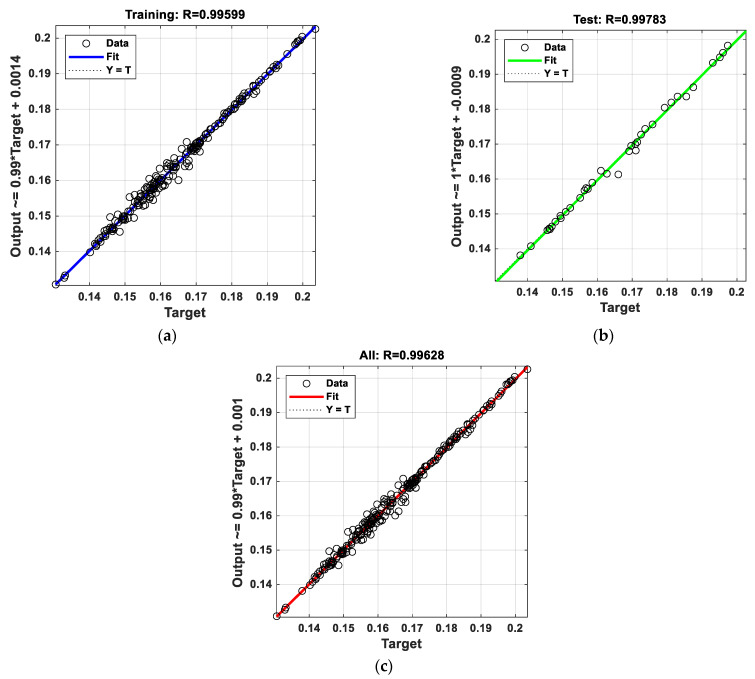
Correlation coefficient (R) plots: (**a**) training, (**b**) test, and (**c**) all.

**Figure 10 micromachines-16-00504-f010:**
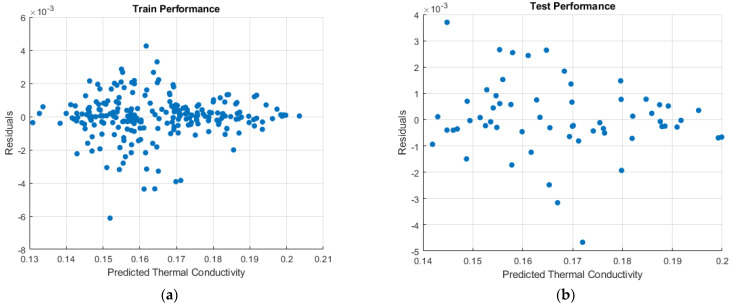
Residual plots: (**a**) train and (**b**) test.

**Figure 11 micromachines-16-00504-f011:**
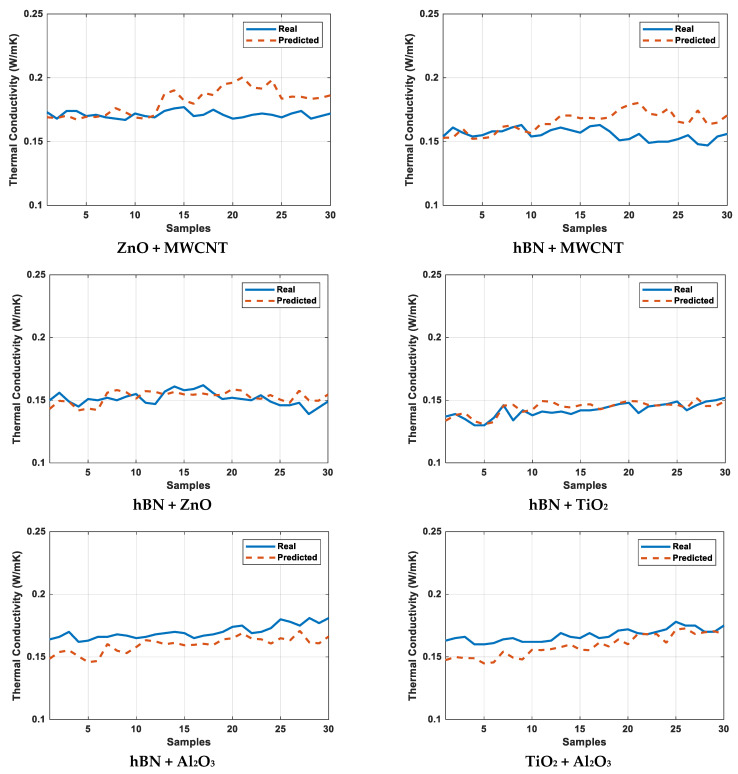
Result plots of calculated actual (measured) and model-predicted thermal conductivity for hybrid mixtures.

**Figure 12 micromachines-16-00504-f012:**
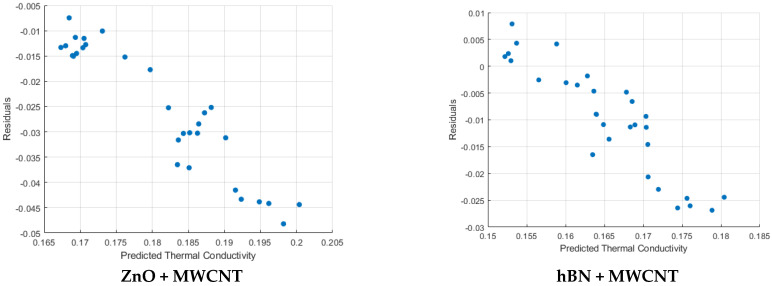
Residual plots for all hybrid nanofluids.

**Figure 13 micromachines-16-00504-f013:**
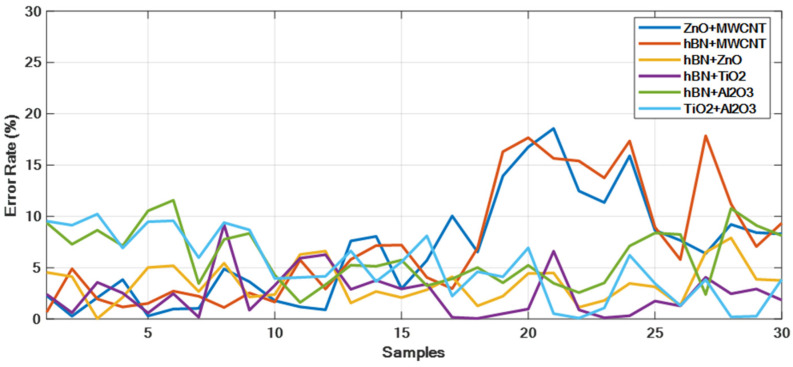
Calculated error rate plots for hybrid mixtures.

**Figure 14 micromachines-16-00504-f014:**
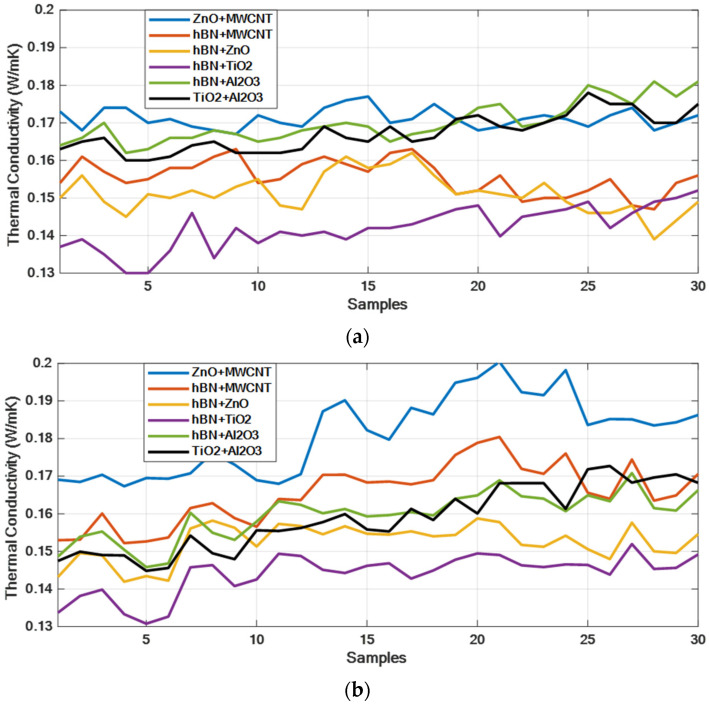
Real (measured) and predicted thermal conductivity results for hybrid mixtures: (**a**) real thermal conductivity results and (**b**) predicted thermal conductivity results.

**Table 1 micromachines-16-00504-t001:** Technical specifications of sunflower oil *.

Oil Type	Density (15 °C, g/mL)	Viscosity (40 °C, mm^2^/s)	Flash Point (°C)	Appearance	Additions (%)
Sunflower oil	0.888	34.25	130	Clear, light yellow	Stabilizer: 0.3 Antifoam: 0.0015

* Information was obtained from the manufacturer.

**Table 2 micromachines-16-00504-t002:** Properties of the nanoparticles used *.

Nanoparticle Type	Density (g/cm^3^)	Particle Size (nm)	Purity (%)	Color
Hexagonal boron nitride (hBN)	2.29	65–75	99.8	White
Çinko Oksit (ZnO)	5.61	18	99.9	White
Multi-walled carbon nanotube (MWCNT)	2.40	48–78	96.0	Black
Titanium dioxide (TiO_2_)	3.90	10–25	99.5	White
Aluminum oxide (Al_2_O_3_)	3.89	13	99.5	White

* Information was obtained from the manufacturer.

**Table 3 micromachines-16-00504-t003:** Formulas used to calculate masses corresponding to volumetric concentrations.

Nanoparticle Volumetric Additive Rate *ϕ* (%)	Nanofluid Volume ∀*n* (mL)	Base Fluid Density ρ*_b_* (kg/m^3^)	Nanoparticle Density ρ*_p_* (kg/m^3^)	Total Nanofluid Mass (For Verification) *m_nf_ = m_np_ + m_bf_ + m_SDS_* (g)
Nanoparticle volume	Base fluid volume	Nanoparticle mass	Base fluid mass	Mass contribution rate
∀_p_ = ϕ∀_n_ (mL)	∀_b_ = ∀_n_ − ∀_p_ (mL)	$m_{p}=\rho$_p_∀_p_ (g)	$m_{b}=\rho$_b_∀_b_ (g)	ϕ_w_ = m_p_/(m_p_ + m_b_) (%)

**Table 4 micromachines-16-00504-t004:** Temperature and thermal conductivity values of mono nanofluids.

Measurement No.	Pure Oil		hBN		ZnO		MWCNT		TiO_2_		Al_2_O_3_	
	T (°C)	k (W/mK)	T (°C)	k (W/mK)	T (°C)	k (W/mK)	T (°C)	k (W/mK)	T (°C)	k (W/mK)	T (°C)	k (W/mK)
1	31.16	0.116	32.28	0.135	35.06	0.161	30.53	0.173	29.84	0.133	35.48	0.168
2	31.29	0.118	31.96	0.142	34.78	0.165	30.65	0.170	30.02	0.131	35.62	0.171
3	30.98	0.112	32.15	0.145	34.95	0.159	30.48	0.176	29.75	0.132	34.95	0.170
4	31.18	0.114	32.65	0.135	35.16	0.159	30.61	0.172	29.80	0.130	35.27	0.170
5	31.56	0.117	32.54	0.132	35.26	0.162	30.50	0.173	29.90	0.128	35.98	0.165
6	31.15	0.114	32.33	0.134	35.14	0.160	30.48	0.174	29.84	0.131	35.79	0.166
7	39.28	0.131	44.75	0.152	39.92	0.164	41.36	0.174	40.36	0.137	40.75	0.172
8	40.05	0.124	45.01	0.150	40.07	0.171	41.45	0.178	40.40	0.140	41.02	0.165
9	39.05	0.136	45.00	0.148	39.86	0.170	41.38	0.174	40.42	0.129	41.02	0.164
10	38.97	0.134	44.67	0.146	39.74	0.163	41.30	0.172	40.36	0.135	40.72	0.174
11	39.14	0.127	44.45	0.156	40.15	0.160	41.29	0.173	40.20	0.142	40.65	0.172
12	39.78	0.128	44.66	0.154	39.97	0.162	41.34	0.175	40.34	0.142	40.90	0.173
13	51.59	0.132	49.51	0.147	47.48	0.170	49.89	0.194	49.75	0.140	51.51	0.179
14	51.56	0.141	49.56	0.145	47.58	0.175	50.01	0.196	50.20	0.142	52.03	0.182
15	50.99	0.129	49.15	0.149	47.12	0.168	49.85	0.189	50.10	0.140	51.85	0.176
16	52.07	0.131	49.83	0.151	47.63	0.165	49.78	0.188	49.50	0.139	50.95	0.175
17	51.58	0.132	49.06	0.142	48.02	0.174	49.79	0.194	49.45	0.145	50.99	0.182
18	51.48	0.136	49.82	0.146	47.25	0.170	49.95	0.193	49.50	0.142	51.49	0.179
19	63.28	0.127	61.03	0.148	56.49	0.172	57.59	0.205	60.15	0.148	59.85	0.188
20	63.18	0.136	61.54	0.152	56.19	0.175	57.84	0.205	60.22	0.145	60.09	0.185
21	63.45	0.120	60.85	0.150	56.78	0.176	57.48	0.210	60.02	0.149	60.07	0.193
22	63.98	0.121	60.56	0.145	55.87	0.170	57.62	0.203	59.95	0.150	59.48	0.192
23	62.98	0.134	61.24	0.144	56.01	0.170	57.60	0.202	60.35	0.150	59.71	0.192
24	63.06	0.134	61.00	0.146	56.09	0.174	57.61	0.208	60.20	0.147	59.87	0.185
25	68.87	0.138	65.85	0.144	66.85	0.168	65.79	0.193	68.75	0.152	70.63	0.192
26	68.91	0.141	65.27	0.139	67.01	0.167	65.87	0.196	69.13	0.150	70.51	0.195
27	68.84	0.129	65.81	0.154	67.09	0.168	65.52	0.195	69.02	0.149	70.48	0.190
28	68.79	0.127	65.69	0.142	66.95	0.169	65.58	0.192	68.75	0.152	70.60	0.189
29	68.74	0.131	65.80	0.142	66.75	0.168	65.90	0.194	68.85	0.154	70.55	0.188
30	69.10	0.138	65.75	0.149	65.23	0.170	65.95	0.195	68.92	0.150	70.40	0.189

**Table 5 micromachines-16-00504-t005:** Temperature and thermal conductivity values of hybrid nanofluids.

Measurement No.	ZnO + MWCNT		hBN + MWCNT		hBN + ZnO		hBN + TiO_2_		hBN + Al_2_O_3_		TiO_2_ + Al_2_O_3_	
	T (°C)	K (W/mK)	T (°C)	k (W/mK)	T (°C)	k (W/mK)	T (°C)	k (W/mK)	T (°C)	k (W/mK)	T (°C)	k (W/mK)
1	35.31	0.173	30.60	0.154	36.75	0.150	32.91	0.137	33.05	0.164	30.41	0.163
2	35.42	0.168	30.62	0.161	36.58	0.156	32.65	0.139	33.50	0.166	30.55	0.165
3	35.51	0.174	30.59	0.157	36.85	0.149	33.15	0.135	33.13	0.170	30.45	0.166
4	35.18	0.174	30.54	0.154	36.65	0.145	32.95	0.130	33.25	0.162	30.36	0.160
5	35.24	0.170	30.69	0.155	36.66	0.151	32.54	0.130	32.85	0.163	30.25	0.160
6	35.15	0.171	30.62	0.158	36.95	0.150	33.15	0.136	33.01	0.166	30.40	0.161
7	41.89	0.169	38.49	0.158	42.79	0.152	39.64	0.146	40.46	0.166	40.97	0.164
8	41.92	0.168	38.62	0.161	41.98	0.150	39.75	0.134	40.95	0.168	41.08	0.165
9	41.86	0.167	38.54	0.163	43.05	0.153	40.10	0.142	39.98	0.167	40.78	0.162
10	41.87	0.172	38.41	0.154	43.10	0.155	39.55	0.138	40.39	0.165	40.69	0.162
11	41.66	0.170	38.42	0.155	42.85	0.148	39.45	0.141	40.45	0.166	40.98	0.162
12	41.94	0.169	38.53	0.159	43.00	0.147	39.41	0.140	40.44	0.168	41.19	0.163
13	55.54	0.174	48.66	0.161	52.29	0.157	48.18	0.141	52.67	0.169	51.85	0.169
14	55.56	0.176	48.63	0.159	52.31	0.161	48.08	0.139	52.80	0.170	52.00	0.166
15	55.47	0.177	48.59	0.157	52.46	0.158	47.68	0.142	52.74	0.169	52.13	0.165
16	55.68	0.170	48.71	0.162	52.06	0.159	48.15	0.142	52.64	0.165	51.65	0.169
17	55.49	0.171	48.75	0.163	52.15	0.162	48.32	0.143	52.39	0.167	51.54	0.165
18	55.44	0.175	48.56	0.158	52.46	0.156	48.07	0.145	52.61	0.168	51.99	0.166
19	64.04	0.171	57.57	0.151	57.96	0.151	56.01	0.147	61.41	0.170	60.82	0.171
20	64.15	0.168	57.84	0.152	58.03	0.152	55.92	0.148	61.25	0.174	60.85	0.172
21	63.89	0.169	57.64	0.156	58.04	0.151	56.03	0.140	61.58	0.175	60.51	0.169
22	63.98	0.171	57.29	0.149	57.80	0.150	55.75	0.145	61.45	0.169	60.72	0.168
23	64.07	0.172	57.41	0.150	58.00	0.154	55.80	0.146	61.56	0.170	60.86	0.170
24	64.10	0.171	57.69	0.150	57.99	0.149	55.98	0.147	61.23	0.173	61.26	0.172
25	72.92	0.169	67.88	0.152	72.30	0.146	69.12	0.149	68.46	0.180	70.07	0.178
26	73.01	0.172	67.89	0.155	71.95	0.146	69.03	0.142	68.37	0.178	70.15	0.175
27	73.03	0.174	67.91	0.148	72.05	0.148	69.00	0.146	69.02	0.175	69.85	0.175
28	72.80	0.168	67.67	0.147	72.95	0.139	68.50	0.149	68.24	0.181	69.95	0.170
29	72.81	0.170	67.73	0.154	72.35	0.144	68.75	0.150	68.34	0.177	70.20	0.170
30	72.95	0.172	67.89	0.156	72.22	0.149	69.71	0.152	68.32	0.181	70.10	0.175

**Table 6 micromachines-16-00504-t006:** Training parameters for 1000 epochs.

Parameter	Initial Value	Stopped Value	Target Value
Epoch	0	1000	1000
Elapsed time	-	00:00:10	-
Performance	0.00494	1.53 × 10^−6^	0
Gradient	0.016	1.75 × 10^−7^	10^−7^
Mu	0.005	0.05	10^10^
Effective # Param	161	43.6	0
Sum Squared Param	175	22.6	0

**Table 7 micromachines-16-00504-t007:** General results of our model.

		MAE	MSE	MAPE	R^2^
**Random**	**Test (20%)**	**0.0010**	**1.76 × 10^−6^**	**0.1490**	**0.9999**
	**Train (80%)**	0.0005	8.55 × 10^−7^	0.0077	1
**Fold-1**	**Test**	0.0010	2.00 × 10^−6^	0.6448	0.9909
	**Train**	0.0010	2.00 × 10^−6^	0.5223	0.9926
**Fold-2**	**Test**	0.0009	2.00 × 10^−6^	0.5596	0.9907
	**Train**	0.0008	1.00 × 10^−6^	0.4815	0.9940
**Fold-3**	**Test**	0.0010	2.00 × 10^−6^	0.6488	0.9907
	**Train**	0.0009	2.00 × 10^−6^	0.5375	0.9925
**Fold-4**	**Test**	0.0012	3.00 × 10^−6^	0.7657	0.9845
	**Train**	0.0009	1.00 × 10^−6^	0.5042	0.9937
**Fold-5**	**Test**	0.0012	3.00 × 10^−6^	0.7462	0.9885
	**Train**	0.0008	1.00 × 10^−6^	0.4763	0.9941
**Average 5-fold**	**All**	0.0011	2.52 × 10^−6^	0.6677	0.9883

**Table 8 micromachines-16-00504-t008:** Performance results.

	ZnO + MWCNT	hBN + MWCNT	hBN + ZnO	hBN + TiO_2_	hBN + Al_2_O_3_	TiO_2_ + Al_2_O_3_
**MAE**	0.0246	0.0113	0.0061	0.00117	0.0088	0.0085
**MSE**	0.0008	0.0002	0.0001	0.0002	0.0001	0.0001
**MAPE**	15.64	6.13	2.05	7.44	0.52	2.28
**R^2^**	0.9693	0.9925	0.9977	0.9919	0.9963	0.9932

**Table 9 micromachines-16-00504-t009:** Statistical error rates.

	ZnO + MWCNT	hBN + MWCNT	hBN + ZnO	hBN + TiO_2_	hBN + Al_2_O_3_	TiO_2_ + Al_2_O_3_
Max	18.5711	17.8538	7.9092	9.2131	11.577	10.2272
Mean	6.7254	7.3220	3.5107	2.4979	6.1420	5.12931
Min	0.2788	0.6692	0.0439	0.0586	1.6121	0.08751
Std	5.1752	5.7071	1.8855	2.2245	2.7797	3.20273

**Table 10 micromachines-16-00504-t010:** Performance metrics for different scenarios.

Number of Hidden Layers	FeedForwardNet		MAE	MSE	MAPE	R2
2	[5 10]	Train	0.000996	2.17 × 10^−6^	0.15876012	0.9999
		Test	0.000721	1.17 × 10^−6^	0.00235251	**1**
	[10 5]	Train	0.000945	1.67 × 10^−6^	0.00244054	0.9999
		Test	0.000700	1.0 × 10^−6^	0.00602084	1.0000
	[10 10]	Train	0.0010	**1.76 × 10^−6^**	**0.1490**	0.9999
		Test	0.0005	**8.55 × 10^−7^**	**0.0077**	**1**
	[20 20]	Train	0.001188	3.62 × 10^−6^	0.08861923	0.9999
		Test	**0.000480**	9.4 × 10^−7^	0.02950661	1.0000
3	[5 5 5]	Train	**0.000900**	3.18 × 10^−6^	0.04344342	0.9999
		Test	0.000716	1.74 × 10^−6^	0.04261764	0.9999
	[10 5 10]	Train	0.000954	2.10 × 10^−6^	0.07332192	0.9999
		Test	0.000636	9.7 × 10^−7^	0.00020328	**1**
	[10 10 10]	Train	0.013284	2.5 × 10^−4^	2.38267938	0.9899
		Test	0.012206	2.18 × 10^−4^	0.72491018	0.9916
	[20 20 20]	Train	0.013867	2.6 × 10^−4^	1.74978362	0.9899
		Test	0.012032	2.1 × 10^−4^	1.21135047	0.9915

**Table 11 micromachines-16-00504-t011:** Comparison with similar studies conducted in recent years.

Study, Year	Material/Method		MAE/MAD	MAPE	MSE/RMSE	R^2^	R
Yunyan Shang et al. [38], 2024	MXene/graphene	GS-MLPNN	-	0.5261	0.000027	0.99882	0.99941
		RS-MLPNN	-	0.6046	0.000055	0.99774	0.99887
		Bayesian-MLPNN	-	3.1981	0.00087	0.96234	0.98099
Sahin et al. [39], 2024	Fe_3_O_4_-MWCN/water	GMDH + NSGA II	0.0009/-	** 0.125 **	7.8 × 10^−6^	1	0.9954
		GMDH + MOWOA	0.000915/-	0.1285	7.89 × 10^−6^	0.999998	0.99537
		GMDH + MOMFO	0.00093/-	0.1292	8.2 × 10^−6^	0.999997	0.99511
Praveen Kumar Kanti et al. [40], 2024	GO + TiO_2_/GO + SiO_2_ (Test data)	Random Forest	-	3.93	0.0052	0.9405	-
		Gradient Boost	-	4.17	0.0056	0.9366	-
		Decision Tree	-	4.54	0.0069	0.9217	-
M. Dinesh Babu et al. [41], 2025	Al_2_O_3_-CuO/water	Levenberg–Marquardt + ANN			0.0023		0.9999
Shekhar et al. [42], 2025	Al_2_O_3_, CeO_2_, and CuO	GBR-GSO	-/0.0480	-	-/0.00157	-	0.9995
Fevzi Sahin [43], 2025	Al_2_O_3_/ SiO_2_	Levenberg–Marquardt + MLP	-	-	8.2175 × 10^−5^	-	0.99958
Our proposed method	ZnO + MWCNT, hBN + MWCNT, hBN + ZnO, hBN + TiO_2_, hBN + Al_2_O_3_ ve TiO_2_ + Al_2_O_3_	Bayesian + FFANN	0.0010	**0.1490**	**1.76 × 10^−6^**	0.9999	0.9962

## Data Availability

Data are available upon request from the authors.

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
