# Peer review of "Predicting Thermal Conductivity of Nanoparticle-Doped Cutting Fluid Oils Using Feedforward Artificial Neural Networks (FFANN)"

_micromachines, 2025, doi:10.3390/mi16050504_

Round 1

Reviewer 1 Report

Comments and Suggestions for Authors

SEE THE ATTAACHED FILE

Comments on the Quality of English Language

The quality is good

Reviewer 2 Report

Comments and Suggestions for Authors

This study explores the effect of thermal conductivity, influenced by the densities of mono and hybrid nanofluids. Nanofluids were created by dispersing hexagonal boron nitride (hBN), Zinc Oxide (ZnO), Multi-Walled Carbon Nanotubes (MWCNT), Titanium Dioxide (TiOâ‚‚), and Aluminum Oxide (Alâ‚‚O₃) nanoparticles in sunflower oil. Hybrid nanofluids combined two nanoparticles, such as ZnO+MWCNT and hBN+ZnO. A dataset of 180 data points was collected by measuring the thermal conductivity and density of these nanofluids at temperatures from 30°C to 70°C.One thing is the methods, in this case more direct and with interest, and the other is that no work can miss the theoretical bases and fundamentals. I see the modern view of new cutting fluids is not update, for instance the idea of using simultaenousoly cryogenics is not defined, please see Drilling of CFRP-Ti6Al4V stacks using CO2-cryogenic cooling, Journal of Manufacturing Processes 64, 58-66 in which a discussion is open, or parametric Optimization and Effect of Nano-Graphene Mixed Dielectric Fluid on Performance of Wire Electrical Discharge Machining Process of Ni55.8Ti Shape …, Materials 14 (10), 2533 in other filed, but with open to use nano particles.

Regarding the ANN…this is OK, but how to link AI new tools with mechanical aspects requires to define the pair of works on which the approach is based, as it was for instance: Using artificial neural networks for the prediction of dimensional error on inclined surfaces manufactured by ball-end milling, The International Journal of Advanced Manufacturing Technology 83, 847-859 and Effects of tool deflection in the high-speed milling of inclined surfaces, The International Journal of Advanced Manufacturing Technology 24, 621-631  se that one is the technique phiscally studied and the other is the data model.

Eliminate Importance of the study…reader will say that. 1.3 must be reduce to 1 paragraph.

You use 7 pages for introduction, missing above works and with much unnecessary information.

Sunflower oil, please for defining it take care of deep studies, as it was: Sustainability analysis of lubricant oils for minimum quantity lubrication based on their tribo-rheological performance, Journal of Cleaner Production 164, 1419-1429  See that they show tribology aspects as well.

Some particles are harmful for workers, please discuss it. ELIMINATE FIGURE 3

Conclusions: give 4-5 more defined, with values.

Paper is a fresh view, but it needs above changes.

In addition, refs 13-16 can be replace by better ones. See for instance In pursuit of sustainable cutting fluid strategy for machining Ti-6Al-4V using life cycle analysis, Sustainable Materials and Technologies 29, e00301 that include a LCA analysis.

Please define better the basis and fundamentals.

Round 2

Reviewer 1 Report

Comments and Suggestions for Authors

The manuscript has been substantially revised in several sections; however, several issues still remain and need to be addressed to improve the overall clarity and scientific rigor of the work before the publishing in the journal

Round 3

Reviewer 1 Report

Comments and Suggestions for Authors

All the suggestions have been satisfied so that the manuscript can be published.